# Dietary–Physical Activity Patterns in the Health Context of Older Polish Adults: The ‘ABC of Healthy Eating’ Project

**DOI:** 10.3390/nu14183757

**Published:** 2022-09-11

**Authors:** Marta Jeruszka-Bielak, Jadwiga Hamulka, Ewa Czarniecka-Skubina, Monika Hoffmann, Eliza Kostyra, Beata Stasiewicz, Jan Jeszka, Lidia Wadolowska

**Affiliations:** 1Department of Human Nutrition, Institute of Human Nutrition Sciences, Warsaw University of Life Sciences (WULS-SGGW), Nowoursynowska 159C, 02-776 Warsaw, Poland; 2Department of Food Gastronomy and Food Hygiene, Institute of Human Nutrition Sciences, Warsaw University of Life Sciences (WULS), 159C, 02-776 Warsaw, Poland; 3Department of Functional and Organic Food, Institute of Human Nutrition Sciences, Warsaw University of Life Sciences (SGGW-WULS), 159C, 02-776 Warsaw, Poland; 4Department of Human Nutrition, Faculty of Food Sciences, University of Warmia and Mazury in Olsztyn, Sloneczna 45F, 10-718 Olsztyn, Poland; 5Department of Human Nutrition and Dietetics, Poznań University of Life Sciences, Wojska Polskiego 31, 60-624 Poznan, Poland

**Keywords:** older adults, food consumption, physical activity, dietary patterns, Mini Nutritional Assessment (MNA^®^), functional limitations

## Abstract

The study aimed to analyze the dietary–physical activity patterns (D-PAPs) in the health context of Polish people aged 60+ years. A total of 418 respondents across Poland were recruited; however, the final analysis included 361 women and men aged 60–89 years old. D-PAPs were derived using a principal component analysis (PCA); input variables were the frequency of consumption of 10 food groups and physical activity. Finally, three D-PAPs were identified: ‘Pro-healthy eating and more-active’, ‘Sweets, fried foods and sweetened beverages’, and ‘Juices, fish and sweetened beverages’. We developed the Functional Limitations Score (FLS) using the Mini Nutritional Assessment (MNA^®^). A logistic regression was applied to verify the association between the D-PAPs and health-condition outcomes. Older adults were more likely to adhere to the upper tertile of the ‘Pro-healthy eating and more-active’ pattern, with good/better self-reported health status in comparison with their peers (OR = 1.86) or with good/very good self-assessed appetite (OR = 2.56), while this was less likely for older adults with malnutrition risk (OR = 0.37) or with a decrease in food intake (OR = 0.46). Subjects with a decrease in food intake (OR = 0.43), who declared a recent weight loss (OR = 0.49), or older adults in the upper tertile of the FLS (OR = 0.34) were less likely to adhere to the upper tertile of the ‘Sweets, fried foods and sweetened beverages’ pattern. The decrease in food intake due to a loss of appetite or chewing or swallowing difficulties was inversely associated with the ‘Pro-healthy eating and more-active’ pattern characterized by a relatively high frequency of consumption of vegetables, fruit, water, dairy, and grains and a high physical activity. In the interest of the good nutritional status and health of older adults, special attention should be paid to removing limitations in meal consumption, including improved appetite.

## 1. Introduction

Aging of societies, which has become a global problem in recent decades, has a significant impact on many areas of the functioning of states, including public health. Favorable physical, mental, and social health of older people contributes to their greater independence, which in turns results in a reduced economic burden and involvement of states in supporting older adults. These are also important from the individual perspective, as they increase the life quality and years of healthy life [1,2].

According to Lalonde’s Health Field Concept, lifestyle is one out of four fields that affect health, which includes, among others, nutrition and physical activity [3]. Moreover, improving only three lifestyle behaviors: dietary habits, physical activity, and tobacco use, could prevent 80% of coronary heart disease and type 2 diabetes, as well as 40% of cancers [4]. It was emphasized that eating habits may become the most important modifiable risk factor in men and women and in all parts of the world [4,5].

A large body of literature shows that the most common age-associated diseases, including cardiovascular diseases, particular cancers, and type 2 diabetes, are related to the ‘Western diet’, which is high in saturated and trans fatty acids, cholesterol, free sugars, and salt but low in dietary fiber, vitamins, and minerals—specifically, antioxidants—due to a lack of vegetables and fruit, legumes, whole grains, and fish. In contrast, a number of dietary patterns such as the DASH Diet, Mediterranean diet, Okinawan Diet, Med Diet-DASH Intervention for Neurodegenerative Delay (MIND), and New Nordic Diet (NND) have been reported to be beneficial in reducing age-related diseases and creating a longer life expectancy [2,4,6,7].

In addition to diet, regular physical activity is advantageous for maintaining physical, mental, and social health, particularly in older adults, and generally facilitates healthy aging [8]. The WHO recommendations for older adults include at least 150–300 min of moderate-intensity aerobic physical activity weekly (or at least 75–150 min of vigorous-intensity aerobic physical activity or a combination of both) with a special focus on functional balance and strength training at least three times a week to enhance functional capacity and to prevent falls [9].

An unbalanced diet and reduced physical activity are often reported in older adults [10,11,12]; however, the determinants of these unhealthy behaviors are still under investigation, as they are complex and multifactorial [12,13]. Dietary intake and food choices are influenced by a broad range of factors that operate on individual, interpersonal, community, and policy levels. In older adults, special attention is paid to health-related factors that include chemosensory changes, chewing and swallowing difficulties, and chronic diseases and medication use, as well as psychological factors such as cognitive impairments or depression [10,12]. Nevertheless, which factors and to what extent they affect the dietary behaviors of older people are not fully recognized yet.

It is worth noting that in the nutrition field, the interest of the scientific community has recently shifted from analyzing nutrient intakes to whole-food consumption and finally to dietary patterns; the last approach enables researchers to examine the overall diet, which may have some advantages [2,14]. Dietary patterns represent a broader picture of food and nutrient consumption, can capture the complexity of the diet and interactive effects of varied food groups or nutrients, and can be more predictive of disease risk. Moreover, they can provide information about the following (or not following) of the current dietary guidelines by populations of interest [14,15]. Nevertheless, there is little research that explored the dietary patterns combined with physical activity and their relationships with health and socioeconomic status in older populations [16]. Taking the above into account, the study aimed to analyze the dietary–physical activity patterns (D-PAPs) in the health context of Polish people aged 60+ years.

## 2. Materials and Methods

### 2.1. Study Design and Participants

The present study was a part of the national multicenter ‘ABC of Healthy Eating’ project; detailed information on the methodology and procedures applied within this project (first edition; older adults) was published previously [17]. Briefly, the study was designed as cross-sectional and was conducted in 8 locations covering the entire territory of Poland in 2015 (Appendix A). Inclusion and exclusion criteria are presented in Figure 1. The final study sample consisted of 361 women and men aged 60–89 years old.

Respondents 60+ years old were invited to participate in this project. Recruitment was carried out in urban, suburban, and rural areas through universities of the 3rd century, senior houses of daily living, communal centers, and rural housewife circles’ press advertisements, as well as researchers’ personal contacts. The main inclusion criteria were as follows: age ≥ 60 years, location up to 50 km from the academic centers, no communication problems, and written consent to participate in the study (Figure 1).

Initially, 756 respondents across Poland, including those from urban, suburban, and rural areas, registered for the project, which also included participation in five nutritional workshops. After the initial verification (see inclusion criteria) and sending of the detailed information on the nutritional workshops, 418 participants remained in the study. Then, 57 participants were excluded from analyses because they were less than 60 years old or there was a lack of data on socioeconomic status or food consumption (Figure 1). The final study sample consisted of 361 women and men aged 60–89 years old. Details on the total sample characteristics are given in the Results section.

### 2.2. Ethics Approval

The project followed the ethical standards recognized by the Declaration of Helsinki and was approved by the Bioethics Committee of the Faculty of Medical Sciences, University of Warmia and Mazury in Olsztyn, on 17 June 2010 (Resolution No. 20/2010). Written and informed consent to participate in the study was obtained from all subjects.

### 2.3. Data Collection

A broad range of data were collected and validated questionnaires were used. Firstly, nutritional and health status were assessed using the Mini Nutritional Assessment (MNA^®^) [18], while data on nutritional risk and perceived appetite were assessed with the Simplified Nutritional Appetite Questionnaire (SNAQ) [19,20]. Data on food frequency and socioeconomic issues were collected using a questionnaire based on the Habits and Nutrition Beliefs Questionnaire (KomPAN^®^) [21]. Physical activity was evaluated using the Minnesota Leisure Time Activity Questionnaire (6-item short version) [22]. All questionnaires were self-administered by the participants while researchers supervised the process, explained any uncertainties orally, or helped to fill out the questionnaires when such a need was reported.

The anthropometric measurements such as body weight and height and mid-arm and calf circumferences were taken by well-trained researchers using standard procedures [23,24]. Professional equipment and measuring tape were used (the same type across all the research centers): for measuring weight—a SECA 799 electronic digital scale was used; for height, we used a SECA 220 portable stadiometer. Measurements were taken twice in light clothing and without shoes, and average values were calculated [25].

### 2.4. Dietary–Physical Activity Patterns (D-PAPs)

The Dietary–physical activity patterns (D-PAPs) were derived a posteriori using a principal component analysis (PCA) with a varimax rotation. The input variables were: the frequency of consumption of 10 food groups: fruits, vegetables, dairy, grains, sweets, fried foods, fish, water, juices, and sweetened beverages, as well as the physical activity data. Data on the frequency of food consumption (7 categories) and physical activity (6 categories) were standardized after assigning the appropriate values as given below in Table 1. The sample size was sufficient to derive the D-PAPs, as the ratio of respondents to input variables was 33:1 (361/11) [26,27].

Three PCA-derived dietary–physical activity patterns were identified (Figure 2). The ‘Pro-healthy eating and more-active’ pattern was loaded heavily by the frequent consumption of vegetables, fruits, dairy, and grains, as well as the frequent drinking of water and high physical activity. The ‘Sweets, fried foods and sweetened beverages’ D-PAP reflected mainly the consumption of sweets, fried foods, and sweetened beverages. The consumption of juices, fish, and sweetened beverages contributed heavily to the third pattern, which was labeled ‘Juices, fish and sweetened beverages’. All patterns explained 47% of the total variance; the share in variance explanation equaled 21%, 15%, and 11%, respectively, for the first, second, and third pattern. The values of factor loadings for all three patterns are given in Figure 2. For further analyses, tertile intervals were calculated for each of the PCA-derived D-PAPs. The frequency of food consumption and physical activity categories according to the tertiles of the D-PAPs are shown in Appendix A.

### 2.5. Health Outcomes

We analyzed the dietary–physical activity patterns in the health context of older adults. The health conditions included numerous data originally derived from the Mini Nutritional Assessment (MNA^®^) [18] and the Simplified Nutritional Appetite Questionnaire (SNAQ) [19,20]. In addition, anthropometric measurements (weight, height, and mid-arm and calf circumferences) were used. The BMI and the Malnutrition Indicator Score were calculated and the Functional Limitations Score (FLS) was developed.

The Malnutrition Indicator Score was calculated according to the Mini Nutritional Assessment (MNA^®^) procedure [18]. Based on the points from the total assessment (max. 30 points), participants were allocated into one out of three categories: malnourished (<17 points), at risk of malnutrition (17–23.5 points), or normal nutritional status (24–30 points).

The Functional Limitations Score (FLS) is an originally created score that describes the limitations in daily life of older adults and focuses on physiological and psychological problems. It was calculated as the sum of the points that were assigned to the categories of nine questions chosen from the MNA^®^. The Functional Limitations Score ranged from 0 to 9 points, where a higher sum of points indicated a higher adherence to the FLS and more difficulties in everyday functioning. For further analyses, the FLS was expressed in tertile intervals. The set of components of the FLS and scoring are presented in Table 2.

### 2.6. Socioeconomic Status Index (SESI)

The Socioeconomic Status Index (SESI) included two variables: (i) place of residence and (ii) self-reported economic situation of household. For each component, points were given according to the data presented in Table 3. The SESI score ranged from 0 to 6 points; a higher value indicated a lower socioeconomic situation. Next, tertile intervals for SESI values were calculated and the ‘higher’ (0–1 point), ‘average’ (2 points), and ‘lower’ (3–6 points) socioeconomic status were categorized.

### 2.7. Statistical Analysis

The minimal sample size, which was calculated with regard to the main objective of the project, was described previously [17]. In the current study, the post hoc statistical power was calculated while considering the health outcomes [26,27]. For the data under study, which are shown in the Results section for D-PAPs and health-condition outcomes, when we compared groups with 103–121 respondents per group while assuming a 5% significance level, the statistical power was in the range of 71–98% (with one lower value of 34%) in the comparison of the means of the Malnutrition Indicator Score, the means of the Functional Limitations Score, and the occurrence of risk of malnutrition. For example, when we compared the means (followed by standard deviation in brackets) of the Malnutrition Indicator Score (24.2 (2.9) vs. 25.8 (3.0)), the statistical power was 98%; for the means of the Functional Limitations Score (2.5 (1.6) vs. 2.0 (1.5)), the statistical power was 71%; and for the occurrence of risk of malnutrition (43% vs. 18%), the statistical power was 99%. Thus, we found that the sample size (361), including the size of the groups (103–121 respondents/group), was adequate to interpret the differences between groups if they really existed.

Data are presented as the percentage distribution or means and standard deviations, respectively, for the categorical and continuous data. The differences in socioeconomic and health data within tertile intervals of each dietary–physical activity pattern were examined using Pearson’s chi-squared test with the Yates correction as necessary (categorical variables) or a Kruskal–Wallis test (continuous variables) [28].

A logistic regression analysis was performed. The odds ratios (ORs) and 95% confidence intervals (95% CI) of adherence to the D-PAPs in association with the health-condition outcomes were calculated. The references (OR = 1.00) were the bottom tertiles of each D-PAP and better health-condition outcomes, including the bottom tertile of the Functional Limitations Score, and normal nutritional status. The ORs were adjusted for age (continuous variable in years), gender, BMI (continuous variable in kg/m^2^), and SESI (continuous variable in points). The level of significance was assessed with a Wald’s test [28].

For all analyses, a *p*-value < 0.05 was considered statistically significant. The statistical analyses were performed using STATISTICA software version 12.0 (StatSoft Inc., Tulsa, OK, USA; StatSoft, Krakow, Poland).

## 3. Results

The baseline characteristics of the study sample with regard to the PCA-derived dietary–physical activity patterns are presented in Table 4. Of the 361 participants, 87% were women mostly aged 60–69 years (63%) who were overweight (42%) or obese (43%). There were no significant associations between BMI and D-PAPs. Based on the SESI calculated, 32% of the older adults had a lower socioeconomic status. Significantly, more participants in the bottom tertile compared to the middle or upper tertile of the ‘Pro-healthy eating and more-active’ pattern had a lower socioeconomic status (49 vs. 23%).

### 3.1. Health Outcomes and Dietary–Physical Activity Patterns—Percentage Distribution

The distribution of health outcomes, including the Functional Limitations Score, in tertiles of all three D-PAPs are presented in Table 5.

In the bottom tertile, compared to the middle or upper tertile of the ‘Pro-healthy eating and more-active’ pattern, a higher percentage of participants was at risk of malnutrition; in the upper tertile of FLS, a lower percentage of subjects with very good self-assessed appetite or very good feeling regarding the food taste was found.

In the bottom tertile, compared to the middle or upper tertile of the ‘Sweets, fried foods, sweetened beverages’ pattern, we observed a lower percentage of subjects with an increase in body weight but a higher percentage of those with a very good self-assessed appetite and feeling the taste of food.

In the middle tertile, compared to the bottom or upper tertile of the ‘Juices, fish, sweetened beverages’ pattern, a higher percentage of subjects with a decrease in food intake or who were at risk of malnutrition was reported.

### 3.2. Health Outcomes and Dietary–Physical Activity Patterns—Logistic Regression Analysis

The results of logistic regression analysis are shown in Table 6. Older adults with a self-reported health status as good or better in comparison with other people of the same age (OR = 1.86) or with good or very good self-assessed appetite (OR = 2.56) were more likely to adhere to the upper tertile of the ‘Pro-healthy eating and more-active’ pattern. Older adults that were at risk of malnutrition (OR = 0.37) or with a decrease in food intake in the last 3 months due to loss of appetite, digestive problems, or chewing or swallowing difficulties (OR = 0.46) were less likely to adhere to the upper tertile of this pattern.

Subjects with a decrease in food intake (OR = 0.43), older adults who declared weight loss during the last three months (OR = 0.49), or older adults in the upper tertile of the FLS (OR = 0.34), including those taking more than three prescription drugs per day (OR = 0.54) or having one or more medically certified diseases (OR = 0.39), were less likely to adhere to the upper tertile of the ‘Sweets, fried foods and sweetened beverages’ pattern.

Older adults with psychological stress or acute disease in the past three months (OR = 1.96) or who were at risk of malnutrition (OR = 2.34) were more likely to adhere to the middle tertile of the ‘Juices, fish and sweetened beverages’ pattern. Older adults with good or very good self-assessment appetite (OR = 0.55) or with good or better self-reported health status in comparison with their peers (OR = 0.53) were less likely to adhere to the middle tertile of this pattern. 

## 4. Discussion

A number of findings from our study were noteworthy. Firstly, we observed that the ‘Pro-healthy eating and more-active’ pattern was more frequent among older Polish adults with good or very good self-assessed appetite or who declared their health status as good or better in comparison with their peers. Secondly, the pattern was less frequent among subjects at risk of malnutrition and with a decrease in food intake due to loss of appetite, digestive problems, or chewing or swallowing difficulties. This pattern included higher physical activity; a higher frequency of consumption of healthy foods such as vegetable, fruits, water, dairy, grains; and a lower intake of unhealthy foods such as sweetened beverages. Additionally, a better socioeconomic status was positively related to this pattern, while a higher FLS was negatively related. Moreover, this study provided additional insight into our previous findings for the ‘ABC of Healthy Eating’ project when we analyzed the socioeconomic and eating- and health-related limitations of single-food-group consumption, while herein we focused on dietary patterns derived a posteriori [17]. We also included the physical activity in the analysis because healthier dietary behaviors are often connected with higher physical activity [2], and each of these elements of lifestyle play an important role in the overall health condition, specifically in the older population [29].

Our results confirmed the findings of other research that focused on the association between appetite and dietary behaviors in older adults. A loss of appetite has been related to a decrease in food intake, mainly whole grains, fruits, and vegetables, and general low dietary variety [30]. In a large-scale quantitative study in American older adults, a significantly lower consumption of solid foods, protein-rich foods, grains and whole grains, fruits, and vegetables was found in participants with a poor appetite when compared to those with a very good appetite [31]. Interestingly, in the NU-AGE project conducted among older adults in five European countries, a better self-reported appetite was related to a more positive nutrition-related attitude, including a healthy diet and more conscious/healthier food choices [32].

As the subsequent consequences of a poor appetite encompass a further deterioration of health (e.g., undernutrition, sarcopenia, and frailty) and higher rates of morbidity and mortality, there is an urgent need to counteract the loss of appetite (and development of anorexia of aging) in older people [30,33]. Solutions should reflect the diagnosed reasons, which may be of physiological, psychological, social, or environmental origin [34,35,36]. The proposed approaches include production of innovative foods with improved flavor, texture, and perceived palatability, as well as the provision of dietary variety to older adults and feeding assistance [31,36].

One of the physiological factors involved in the loss of appetite is a decline in the perception of smell and taste due to, e.g., the changes in the number and functionality of taste buds that occur with aging. This may result in a more monotonous diet [37]. Interestingly, we observed that healthier lifestyle patterns could have been associated with self-rating the feeling the taste of food as ‘very good’, although this was only shown in the univariate analysis.

Moreover, a loss of appetite is associated with a number of diseases that are common among older persons, including depression and cognitive impairment [36]. In the present study, we found a lower percentage of older adults with neuropsychological problems in the upper tertile (when compared to the bottom and middle tertiles) of the ‘Pro-healthy eating and more-active’ pattern. Furthermore, a higher FLS was associated with lower adherence to this pattern while a ‘better’ self-rated health status was associated with a higher adherence to this pattern. We speculated that the presence of a higher number of functional limitations, including some neuropsychological problems, and a ‘worse’ perceived health status influenced the feeling of food taste and appetite, which, together with digestive problems or chewing or swallowing difficulties, caused a decrease in food intake in general and in healthy foods in particular, and finally resulted in the worsening of the nutritional status of the older adults. However, the connection between the risk of malnutrition and a lower adherence to a healthy-oriented lifestyle found in this study might have been bidirectional, and a worsened nutritional status may have been the reason and not the consequence of an imbalanced diet observed at the moment of data collection. Moreover, physical activity cannot be neglected; on the one hand, physical inactivity in older adults results in the premature onset of disease and frailty, and on the other hand, it may occur due to worsening of the nutritional and health status [38,39].

We were not surprised at the outcome of this study that a lower adherence to the unhealthy dietary pattern of ‘Sweets, fried foods, sweetened beverages’ was related to a decrease in food intake and a higher Functional Limitations Score, including conditions such as the presence of a medically certified disease and taking more than three prescription drugs/day, as well as higher number of medically certified diseases and a decrease in body weight. It is well documented that a number of diseases and medications reduce food intake, including those foods typically perceived as palatable ones (such as sweets or fried foods), mainly through modification of taste sensation and appetite. Polypharmacy additionally poses a risk due to drug–drug interactions and gastrointestinal problems, a further decline in food intake, and the subsequent weight loss [34,36,40,41].

Our research revealed that the ‘Pro-healthy eating and more-active’ pattern was associated with a better socioeconomic status. This outcome was in agreement with numerous studies that demonstrated the positive impact of a higher socioeconomic status on attitudes toward healthy dietary patterns [32], food choices [42], dietary behaviors [13,43], and nutritional status [44], as well as higher physical activity [38,39] in older adults.

Although we did not note a significant impact of either age or gender on the adherence to the ‘Pro-healthy eating and more-active’ pattern, a slightly higher proportion of younger subjects and women were found in the upper tertile of this pattern. A number of studies showed that women tended to make healthier food choices and had healthier dietary behaviors when compared to men [14], which may have resulted at least partially from their better nutrition-related knowledge [45] and more positive nutrition-related attitudes [32], although the differences between both genders might diminish in the future due to the changes in the traditional roles of women and men in food supply and meal preparation. Regarding age, a similar negative association between aging and diet quality was observed. For example, in a Canadian population, the tendency of the oldest old (aged 85 years and older) when compared to the younger old (aged 65–85 years) to follow a Western dietary pattern (characterized by french fries, red meat, and processed meat) and not a nutrient-rich dietary pattern (which included, among others, fruits, vegetables, and whole grains) was reported [46].

In the present study, three PCA-derived dietary–physical activity patterns involved all input variables, which were the frequency of consumption of 10 food groups. Some foods were frequently consumed within more than one pattern. For example, the consumption of sweetened beverages contributed heavily to the two of three identified patterns, which were labeled ‘Sweets, fried foods and sweetened beverages’ and ‘Juices, fish and sweetened beverages’. This relatively high frequency of sweetened-beverage consumption could be a consequence of the ability to recognize the sweet taste, which is kept even in advanced age [47]. For this reason, older adults often drink sweetened beverages instead of water. Moreover, older adults often drink juices and sweetened beverages in place of fruit and vegetable consumption. These unhealthy food choices could result from the relatively high price of fruit and vegetables, the low availability of these products in low season, or chewing difficulties [48].

It is worth noting that the ‘Pro-healthy eating and more-active’ pattern in our study included a higher frequency of consumption of vegetable and fruits, water, dairy, and grains and a lower frequency of sweetened-beverage consumption. Such a dietary pattern might be equated with the current food-based dietary guidelines formulated in Poland [49]. A considerable number of previous studies underlined the role of healthy eating patterns, including a plant-based diet, in the prevention and treatment of age-related diseases and in lowering the risk of all-cause mortality [2,4,14,50]. The importance of a high consumption of vegetables and fruit was specifically emphasized due to its inverse correlation with the risks of noncommunicable diseases, cardiovascular disease, hypertension, type 2 diabetes, and some cancers, as well as with a reduced cognitive decline, dementia, and depression [2,51,52]. Moreover, the higher physical activity included in this pattern was an additional protective factor for the prevention and treatment of the aforementioned diseases and health conditions [8,29].

### Strengths and Limitations

Our study had several strengths. First, the study included a relatively homogeneous group of older adults aged 60+ years from a multicenter study that was a part of the nationwide ‘ABC of Healthy Eating’ project. Although the sample was not randomly selected, it covered the entire territory of Poland and widely reflected the sociodemographic diversity of Poles; however, it did not provide a good basis for generalizations, as the subpopulation of men was under-representative. Secondly, we used a dietary-pattern approach combined with physical activity, which provided a deeper insight into and a broader picture of older adults’ lifestyles.

However, our study was not without limitations. First, the sample size was relatively small and nonrepresentative at the population level; however, after checking the post hoc statistical power, we found that the sample size was adequate to detect differences between the groups (103–121 respondents/group) if they really existed. Furthermore, the sample size was adequate to derive dietary–physical activity patterns because the ratio of respondents to input variables in the PCA was 33:1 (361/11); this ratio should be at least 10:1 [26,27]. Secondly, our analysis was a cross-sectional study, which precluded assessment of the causal relationship between variables. Third, all indexes, including the socioeconomic status index, were constructed using mainly the subjects’ self-reported data such as the economic situations of their households, which may have been biased due to the subjective estimate. However, both the Mini Nutritional Assessment (MNA^®^) [18] and the Simplified Nutritional Appetite Questionnaire (SNAQ) [19] are validated and widely used questionnaires in studies in older populations [18,19,20,53]. Although we used an innovative, originally created score—the Functional Limitations Score (FLS)—this indicator was not validated. Nevertheless, it described the limitations in everyday life of older adults and focused on physiological and psychological problems, and thus may provide a more complex approach to nutrition research in aging populations worldwide, not just in Poland.

## 5. Conclusions

Most of the functional and health problems were recognized as limitations related to all dietary–physical activity patterns. A decrease in food consumption due to loss of appetite, digestive problems, or chewing or swallowing difficulties was inversely associated with the ‘Pro-healthy eating and more-active’ pattern, which included a relatively high frequency of consumption of vegetables, fruits, water, dairy, and grains, as well as high physical activity, among Polish older adults. In the interests of the good nutritional status and health of older adults, special attention should be paid to removing the limitations in eating meals by improving their appetites as well as promoting their physical activity.

## Figures and Tables

**Figure 1 nutrients-14-03757-f001:**
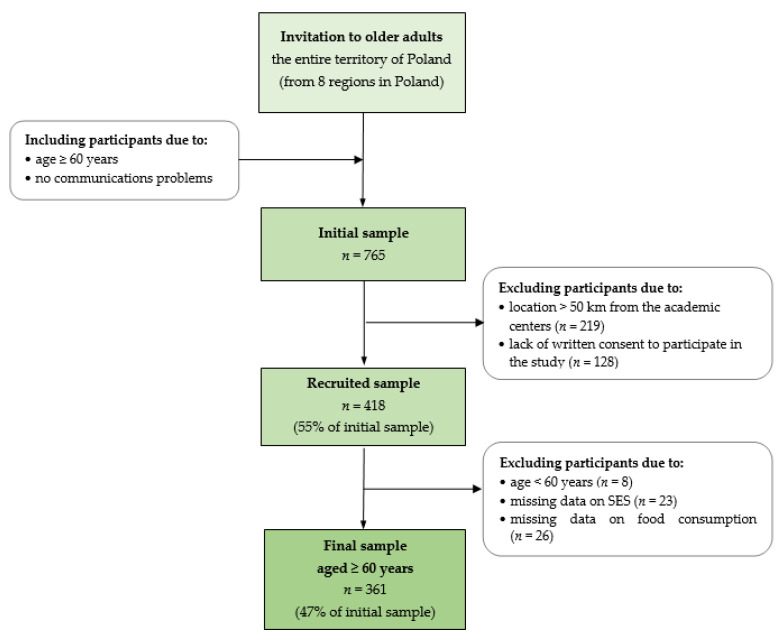
Flowchart of study design and sample collection. Note: SES—socioeconomic status.

**Figure 2 nutrients-14-03757-f002:**
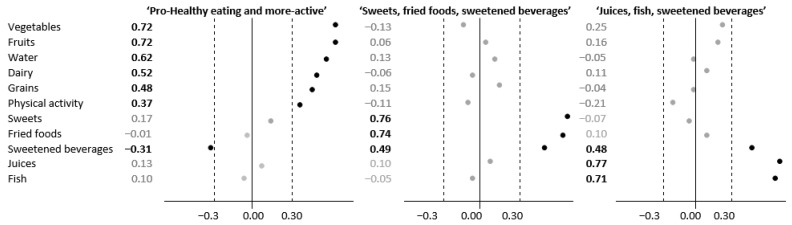
The values of factor loadings for selected food groups and physical activity in PCA-derived dietary–physical activity patterns (D-PAPs) (*n* = 361).

**Table 1 nutrients-14-03757-t001:** Dietary–physical activity patterns (D-PAPs)—component data and scoring.

Components	Scoring (Points)
Food frequency consumption:	
Never or almost never	1
<1 time/week	2
1 time/week	3
2–4 times/week	4
5–6 times/week	5
1 time/day	6
A few times/day	7
Physical activity:	
Practically no physical activity	1
Sedentary lifestyle	2
Light exercise at least 2–4 h/week	3
Moderately intensive exercise for 1–2 h/week or less intensive exercise >4 h/week	4
Moderately intensive exercise >3 h/week	5
Intensive exercise regularly (several times a week)	6

**Table 2 nutrients-14-03757-t002:** Functional Limitations Score (FLS)—component data and scoring.

Components	Scoring (Points)	
	0	1
Dependent life (i.e., in nursing home or hospital)	No	Yes
Limited mobility (i.e., able to get out of bed/chair but does not go out vs. goes out)	No	Yes
Medically certified disease	No	Yes
Psychological stress or acute disease in the last 3 months	No	Yes
Neuropsychological problems	No	Yes
Taking more than 3 prescription medications/day	No	Yes
Pressure sores or skin ulcers	No	Yes
Self-reported health status in comparison with other people of the same age (i.e., weaker/does not know vs. as good/better)	No	Yes
Self-reported nutritional status (malnourished/does not know vs. no nutritional problems)	No	Yes

Range of points: 0–9.

**Table 3 nutrients-14-03757-t003:** The Socioeconomic Status Index (SESI)—component data and scoring.

Components	Scoring (Points)
Place of residence:	
City (>100,000 inhabitants)	0
Town (<100,000 inhabitants)	1
Village	2
Self-reported economic situation of household:	
I live very well—I have enough resources for everything, and I put off savings	0
I live well—I have enough resources for everything, but I don’t put off savings	1
I live thriftily—I have enough resources for everything	2
I live very thriftily—I have enough resources only for basic needs (food/clothing/housing fees)	3
I live poorly—I don’t have enough resources even for basic needs (food/clothing/housing fees)	4

Range of points: 0–6.

**Table 4 nutrients-14-03757-t004:** Baseline characteristics of study sample (sample size and percentage (%) or mean ± SD).

Variables	Total Sample	Dietary–Physical Activity Patterns (Tertiles)											
		‘Pro-Healthy Eating and More-Active’				‘Sweets, Fried Foods, Sweetened Beverages’				‘Juices, Fish, Sweetened Beverages’			
		Bottom	Middle	Upper	*p*-Value	Bottom	Middle	Upper	*p*-Value	Bottom	Middle	Upper	*p*-Value
Sample size (n)	361	121	120	120		121	120	120		120	120	121	
Factor scores of dietary patterns		−2.99 to <−0.36	−0.36 to <0.54	0.54 to 1.92		−2.85 to <−0.40	−0.40 to <0.44	0.44 to 2.98		−2.19 to <−0.53	−0.53 to <0.35	0.35 to 3.03	
Age, years	69.5 ± 5.5	69.5 ± 5.4	70.0 ± 5.6	68.9 ± 5.7	0.299	70.3 ± 5.5 ^a^	68.8 ± 5.5 ^b^	69.3 ± 5.6 ^ab^	0.044	69.1 ± 5.6	70.1 ± 5.8	69.3 ± 5.3	0.400
60–69	229 (63)	74 (61)	72 (60)	83 (69)		65 (54) ^a^	84 (70) ^b^	80 (67) ^b^		75 (62)	74 (62)	80 (66)	
70–89	132 (37)	47 (39)	48 (40)	37 (31)	0.275	56 (46) ^a^	36 (30) ^b^	40 (33) ^b^	0.021	45 (38)	46 (38)	41 (34)	0.748
Gender													
Male	48 (13)	21 (17) ^a^	17 (14) ^ab^	10 (8) ^b^	0.112	16 (13)	16 (13)	16 (13)	1.000	10 (8) ^a^	21 (18) ^b^	17 (14) ^ab^	0.107
Female	313 (87)	100 (83) ^a^	103 (86) ^ab^	110 (92) ^b^		105 (87)	104 (87)	104 (87)		110 (92) ^a^	99 (82) ^b^	104 (86) ^ab^	
BMI (kg/m^2^) ^$^	29.7 ± 4.9	29.6 ± 5.0	30.2 ± 4.9	29.2 ± 4.6	0.321	30.0 ± 4.6	29.5 ± 5.3	29.5 ± 4.7	0.536	29.3 ± 4.7	29.5 ± 5.3	30.2 ± 4.5	0.159
Normal weight, 18.5–24.9	50 (15)	20 (17)	10 (9)	20 (18)		11 (10)	19 (17)	20 (17)		16 (15)	21 (18)	13 (11)	
Overweight, 25.0–29.9	145 (42)	47 (41)	52 (45)	46 (41)	0.307	49 (44)	50 (44)	46 (39)	0.432	48 (45)	49 (43)	48 (40)	0.345
Obesity, ≥30.0	148 (43)	48 (42)	53 (46)	47 (41)		52 (46)	44 (39)	52 (44)		43 (40)	45 (39)	60 (50)	
Place of residence													
Village	41 (11)	21 (17) ^a^	12 (10) ^ab^	8 (7) ^b^		13 (11)	19 (16)	9 (8)		11 (9)	15 (13)	15 (12)	
Town *	27 (7)	9 (7)	11 (9)	7 (6)	0.079	5 (4)	11 (9)	11 (9)	0.120	9 (8)	5 (4) ^a^	13 (11) ^b^	0.324
City **	293 (81)	91 (75)^a^	97 (81)	105 (88) ^b^		103 (85)	90 (75)	100 (83)		100 (83) ^a^	100 (83) ^b^	93 (77) ^ab^	
Self-declared economic situation of household													
I live poorly	23 (6)	14 (12) ^a^	4 (3) ^b^	5 (4) ^b^		7 (6)	7 (6)	9 (8)		7 (6)	7 (6)	9 (7)	
I live very thriftily	66 (18)	31 (26) ^a^	16 (13) ^b^	19 (16) ^b^		22 (18)	22 (18)	22 (18)		19 (16)	27 (23)	20 (17)	
I live thriftily	149 (41)	50 (41)	49 (41)	50 (42)	0.002	44 (36)	59 (49)	46 (38)	0.607	53 (44)	47 (39)	49 (40)	0.907
I live well	70 (19)	18 (15)	25 (21)	27 (23)		28 (23)	18 (15)	24 (20)		24 (20)	20 (17)	26 (21)	
I live very well	53 (15)	8 (7) ^a^	26 (22) ^b^	19 (16) ^b^		20 (17)	14 (12)	19 (16)		17 (14)	19 (16)	17 (14)	
Socioeconomic Status Index (SESI), points	2.1 ± 1.3	2.6 ± 1.4 ^a^	1.9 ± 1.2 ^b^	1.9 ± 1.2 ^b^	<0.001	2.0 ± 1.3	2.3 ± 1.2	2.1 ± 1.3	0.060	2.0 ± 1.3	2.2 ± 1.4	2.2 ± 1.2	0.530
Higher, 0–1	100 (28)	21 (17) ^a^	39 (33) ^b^	40 (33) ^b^		42 (35) ^a^	22 (18) ^b^	36 (30) ^a^		38 (32)	34 (28)	28 (23)	
Average, 2	147 (41)	41 (34)	54 (45)	52 (43)	<0.001	45 (37)	56 (47)	46 (38)	0.070	49 (41)	44 (37)	54 (45)	0.457
Lower, 3–6	114 (32)	59 (49) ^a^	27 (23) ^b^	28 (23) ^b^		34 (28)	42 (35)	38 (32)		33 (28)	42 (35)	39 (32)	

Notes: ^$^ data were obtained for n = 343; * <100,000 inhabitants; ** >100,000 inhabitants; *p*-value—level of significance assessed using a Kruskal–Wallis test (continuous variables) or chi-squared test (categorical variables); ^a, b^ nonidentical superscripts indicate significant difference among tertiles within each pattern, *p* < 0.05; ns—statistically insignificant.

**Table 5 nutrients-14-03757-t005:** Dietary–physical activity patterns (D-PAPs) and health-condition outcomes in older Polish adults (sample size and percentage (%) or mean ± SD).

Variables	Total Sample	Dietary–Physical Activity Patterns (Tertiles)											
		‘Pro-Healthy Eating and More-Active’				‘Sweets, Fried Foods, Sweetened Beverages’				‘Juices, Fish, Sweetened Beverages’			
		Bottom	Middle	Upper	*p*-Value	Bottom	Middle	Upper	*p*-Value	Bottom	Middle	Upper	*p*-Value
Sample size (n)	361/325 ^#^	121/107 ^#^	120/110 ^#^	120/108 ^#^		121/105 ^#^	120/106 ^#^	120/114 ^#^		120/103 ^#^	120/106 ^#^	121/116 ^#^	
Factor scores of dietary patterns		−2.99 to <−0.36	−0.36 to <0.54	0.54 to 1.92		−2.85 to <−0.40	−0.40 to <0.44	0.44 to 2.98		−2.19 to <−0.53	−0.53 to <0.35	0.35 to 3.03	
Malnutrition Indicator Score, points	25.1 ± 2.9	24.2 ± 2.9 ^a^	25.2 ± 2.4 ^b^	25.8 ± 3.0 ^c^	<0.001	24.8 ± 3.2	25.0 ± 2.8	25.4 ± 2.5	0.443	25.6 ± 2.4 ^a^	24.5 ± 2.9 ^b^	25.2 ± 3.1 ^a^	0.014
Nutritional status^1^													
Malnourished, <17.0	3 (1)	1 (1)	0 (0)	2 (2)		1 (1)	1 (1)	1 (1)		0 (0)	0 (0)	3 (3)	
At risk of malnutrition, 17.0–23.5	99 (30)	46 (43) ^a^	33 (30) ^ab^	20 (18) ^b^	0.002	41 (39)	26 (24)	32 (28)	0.218	23 (22) ^a^	44 (42) ^b^	32 (27) ^a^	0.005
Normal nutritional status, 24.0–30.0	223 (69)	60 (56) ^a^	77 (70) ^ab^	86 (80) ^b^		63 (60)	79 (75)	81 (71)		80 (78) ^a^	62 (58) ^b^	81 (70) ^a^	
Weight change ^2^													
No change	208 (58)	70 (58)	67 (56)	71 (59)		65 (54) ^a^	72 (60) ^b^	71 (59) ^b^		74 (62)	70 (58)	64 (53)	
Decreased by more than 3 kg	24 (7)	8 (7)	7 (6)	9 (8)		15 (12) ^a^	6 (5) ^b^	3 (3) ^b^		6 (5)	11 (9)	7 (6)	
Decreased 1–3 kg	42 (12)	17 (14)	12 (10)	13 (11)	0.849	18 (15) ^a^	10 (8) ^b^	14 (12) ^b^	0.027	11 (9)	15 (13)	16 (13)	0.577
Increased	55 (15)	15 (12)	24 (20)	16 (13)		11 (9) ^a^	23 (19) ^b^	21 (18) ^b^		20 (17)	16 (13)	19 (16)	
Does not know	32 (9)	11 (9)	10 (8)	11 (9)		12 (10) ^a^	9 (8) ^b^	11 (9) ^b^		9 (8)	8 (7)	15 (12)	
Self-assessed appetite													
Very weak	7 (2)	4 (3)	2 (2)	1 (1)		2 (2)	0 (0)	5 (4)		2 (2)	1 (1)	4 (3)	
Weak	3 (1)	1 (1)	1 (1)	1 (1)		2 (2)	1 (1)	0 (0)		1 (1)	1 (1)	1 (1)	
Average	94 (26)	42 (35) ^a^	32 (27) ^ab^	20 (17) ^b^	0.030	25 (21) ^a^	39 (33) ^b^	30 (25) ^ab^	0.009	22 (18) ^a^	39 (33) ^b^	33 (27)	0.267
Good	194 (54)	61 (50) ^a^	65 (53) ^ab^	68 (56) ^b^		60 (50) ^a^	67 (56) ^b^	67 (56) ^ab^		68 (57)	63 (53)	63 (52)	
Very good	63 (17)	13 (11) ^a^	20 (17) ^ab^	30 (25) ^b^		32 (26) ^a^	13 (11) ^b^	18 (15) ^ab^		27 (23) ^a^	16 (13) ^b^	20 (17)	
Feeling the taste of food													
Very weak	5 (1)	5 (4)	0 (0)	0 (0)		0 (0)	3 (3)	2 (2)		1 (1)	3 (3)	1 (1)	
Weak	3 (1)	1 (1)	0 (0)	2 (2)		1 (1)	2 (2)	0 (0)		1 (1)	0 (0)	2 (2)	
Average	36 (10)	14 (12) ^a^	14 (12) ^ab^	8 (6) ^b^	0.011	15 (12)	13 (11)	8 (7)	0.007	10 (8)	17 (14)	9 (7)	0.391
Good	212 (59)	75 (62) ^a^	72 (60) ^ab^	65 (54) ^b^		59 (49) ^a^	82 (68) ^b^	71 (59) ^a^		68 (57)	68 (57)	76 (63)	
Very good	105 (29)	26 (21) ^a^	34 (28) ^ab^	45 (38) ^b^		46 (38) ^a^	20 (17) ^b^	39 (33) ^a^		40 (33)	32 (27)	33 (27)	
Decrease in food intake^3^													
Severe decrease in food intake	13 (4)	6 (5)	3 (3)	4 (3)		4 (3)	4 (3)	5 (4)		2 (2) ^ab^	7 (6) ^a^	4 (3) ^b^	
Moderate decrease in food intake	69 (19)	31 (26) ^a^	22 (18) ^ab^	16 (13) ^b^	0.114	32 (26) ^a^	22 (18) ^ab^	15 (13) ^b^	0.103	23 (19) ^ab^	31 (26) ^a^	15 (12) ^b^	0.030
No decrease in food intake	279 (77)	84 (69) ^a^	95 (79) ^ab^	100 (83) ^b^		85 (70) ^a^	94 (78) ^ab^	100 (83) ^b^		95 (79) ^ab^	82 (68) ^a^	102 (84) ^b^	
Functional Limitations Score (FLS), points	2.3 ± 1.5	2.5 ± 1.6	2.2 ± 1.5	2.0 ± 1.5	0.054	2.5 ± 1.6	2.2 ± 1.5	2.1 ± 1.5	0.056	2.1 ± 1.3	2.6 ± 1.6	2.2 ± 1.6	0.051
Bottom, 0	49 (14)	16 (13) ^a^	14 (12) ^b^	19 (16) ^b^		13 (11)	17 (14)	19 (16)		16 (13)	14 (12)	19 (16)	
Middle, 1–2	163 (45)	42 (35) ^a^	61 (51) ^b^	60 (50) ^b^	0.033	46 (38)	59 (49)	58 (48)	0.105	60 (50)	47 (39)	56 (46)	0.279
Upper, 3–9	149 (41)	63 (52) ^a^	45 (38) ^b^	41 (34) ^b^		62 (51)	44 (37)	43 (36)		44 (37)	59 (49)	46 (38)	
FLS components													
Lives dependently ^4^	14 (4)	7 (6)	3 (3)	4 (3)	0.389	6 (5)	4 (3)	4 (3)	0.752	5 (4)	5 (4)	4 (3)	0.923
Limited mobility ^5^	2 (1)	1 (1)	1 (1)	0 (0)	0.606	0 (0)	1 (1)	1 (1)	0.602	0 (0)	1 (1)	1 (1)	0.606
Medically certified disease	223 (62)	73 (60)	70 (58)	80 (67)	0.382	89 (74) ^a^	73 (61) ^b^	61 (51) ^b^	0.001	76 (63)	76 (63)	71 (59)	0.691
Psychological stress or acute disease ^1^	122 (34)	46 (38)	38 (32)	38 (32)	0.484	43 (36)	41 (34)	38 (32)	0.813	31 (26) ^a^	50 (42) ^b^	41 (34) ^ab^	0.035
Neuropsychological problems	57 (16)	23 (19) ^a^	23 (19) ^a^	11 (9) ^b^	0.049	19 (16)	20 (17)	18 (15)	0.939	17 (14)	24 (20)	16 (13)	0.296
Taking more than 3 prescription drugs/day	160 (44)	59 (49)	56 (47)	45 (38)	0.174	67 (55) ^a^	47 (39) ^b^	46 (38) ^b^	0.011	52 (43)	58 (48)	50 (41)	0.530
Pressure sores or skin ulcers	5 (1)	1 (1)	3 (3)	1 (1)	0.441	2 (2)	1 (1)	2 (2)	0.818	0 (0) ^a^	5 (4) ^b^	0 (0) ^a^	0.006
Self-reported health status ^6^:													
Weaker	57 (16)	22 (18)	19 (16)	16 (13)		22 (18)	21 (18)	14 (12)		12 (10)	27 (23)	18 (15)	
Does not know	97 (27)	39 (32)	31 (26)	27 (23)	0.311	32 (26)	29 (24)	36 (30)	0.139	31 (26)	32 (27)	34 (28)	0.097
As good	148 (41)	45 (37)	52 (43)	51 (43)		41 (34)	57 (48)	50 (42)		50 (42)	46 (38)	52 (43)	
Better	59 (16)	15 (12)	18 (15)	26 (22)		26 (21)	13 (11)	20 (17)		27 (23)	15 (13)	17 (14)	
Self-reported nutritional status													
Malnourished/does not know	81 (22)	34 (28)	23 (19)	24 (20)	0.185	24 (20)	29 (24)	28 (23)	0.693	24 (20)	28 (23)	29 (24)	0.731
No nutritional problems	280 (78)	87 (72)	97 (81)	96 (80)		97 (80)	91 (76)	92 (77)		96 (80)	92 (77)	92 (76)	

Notes: ^#^ data for Malnutrition Indicator Score and nutritional status were collected based on the Mini Nutritional Assessment (MNA^®^); ^1^ nutritional status: malnourished (Malnutrition Indicator Score < 17 points), at risk of malnutrition (Malnutrition Indicator Score 17–23.5 points), normal nutritional status (Malnutrition Indicator Score 24–30 points); ^2^ in the last 3 months; ^3^ in the last 3 months due to loss of appetite, digestive problems, chewing or swallowing difficulties; ^4^ in nursing home or hospital; ^5^ is able to get out of bed/chair but does not go out; ^6^ in comparison with other people of the same age; *p*-value—level of significance assessed with Kruskal–Wallis test (continuous variables) or chi-squared test (categorical variables); ^a, b^ nonidentical superscripts indicate significant difference among tertiles within each pattern, *p* < 0.05; ns—statistically insignificant.

**Table 6 nutrients-14-03757-t006:** Odds ratios (ORs with 95% confidence interval (95% CI)) of dietary–physical activity patterns (D-PAPs) according to health-condition outcomes in older Polish adults (*n* = 361).

Health Context		‘Pro-Healthy Eating and More-Active’			‘Sweets, Fried Foods, Sweetened Beverages’			‘Juices, Fish, Sweetened Beverages’		
		Bottom (*n* = 121)	Middle (*n* = 120)	Upper (*n* = 120)	Bottom (*n* = 121)	Middle (*n* = 120)	Upper (*n* = 120)	Bottom (*n* = 120)	Middle (*n* = 120)	Upper (*n* = 121)
Factor scores of dietary patterns		−2.99 to <−0.36	−0.36 to <0.54	0.54 to 1.92	−2.85 to <−0.40	−0.40 to <0.44	0.44 to 2.98	−2.19 to < −0.53	−0.53 to <0.35	0.35 to 3.03
Nutritional status ^1^	Malnourished/at risk of malnutrition (ref. normal)	1	0.54 * (0.30; 0.98)	0.37 ** (0.19; 0.71)	1	0.43 ** (0.23; 0.81)	0.63 (0.36; 1.13)	1	2.34 ** (1.24; 4.41)	1.47 (0.79; 2.73)
Self-assessed appetite	Good/very good (ref. weak/average)	1	1.55 (0.86; 2.79)	2.56 ** (1.35; 4.87)	1	0.62 (0.34; 1.14)	0.77 (0.42; 1.39)	1	0.55 (0.29; 1.01)	0.60 (0.32; 1.11)
Feeling the taste of food	Good/very good (ref. weak/average)	1	1.47 (0.65; 3.34)	1.87 (0.75; 4.66)	1	0.84 (0.37; 1.87)	1.59 (0.67; 3.78)	1	0.52 (0.22; 1.19)	0.91 (0.37; 2.27)
Decrease in food intake ^2^	Yes (ref. no)	1	0.61 (0.32; 1.14)	0.46 * (0.24; 0.91)	1	0.60 (0.32; 1.10)	0.43 ** (0.23; 0.82)	1	1.56 (0.84; 2.89)	0.58 (0.29; 1.16)
Weight change ^3^	Decreased (ref. no change/does not know)	1	0.74 (0.36; 1.54)	0.77 (0.38; 1.57)	1	0.47 * (0.24; 0.95)	0.49 * (0.25; 0.96)	1	1.58 (0.77; 3.24)	1.33 (0.64; 2.76)
	Increased (ref. no change/does not know)	1	1.94 (0.86; 4.39)	1.11 (0.45; 2.74)	1	2.39 * (1.01; 5.67)	2.19 (0.93; 5.20)	1	1.09 (0.50; 2.38)	1.02 (0.48; 2.16)
Functional Limitations Score (tertiles)	Middle (ref. bottom)	1	0.96 (0.40; 2.33)	0.94 (0.41; 2.14)	1	0.69 (0.30; 1.59)	0.91 (0.40; 2.08)	1	1.03 (0.45; 2.35)	0.76 (0.35; 1.63)
	Upper (ref. bottom)	1	0.86 (0.32; 2.34)	0.56 (0.21; 1.52)	1	0.49 (0.18; 1.32)	0.34 * (0.12; 0.98)	1	2.42 (0.84; 7.00)	1.32 (0.47; 3.71)
Self-reported health status ^4^	As good/better (ref. weaker/does not know)	1	1.32 (0.75; 2.32)	1.86 * (1.05; 3.30)	1	1.28 (0.73; 2.52)	0.97 (0.57; 1.66)	1	0.53 * (0.30; 0.93)	0.68 (0.40; 1.18)
Medically certified disease	Yes (ref. no)	1	0.75 (0.43; 1.34)	1.34 (0.75; 2.40)	1	0.56 (0.32; 1.01)	0.39 *** (0.22; 0.68)	1	0.93 (0.53; 1.64)	0.82 (0.47; 1.43)
Psychological stress or acute disease ^3^	Yes (ref. no)	1	0.56 (0.31; 1.02)	0.60 (0.33; 1.07)	1	0.93 (0.53; 1.62)	0.85 (0.48; 1.49)	1	1.96 * (1.10; 3.50)	1.38 (0.77; 2.47)
Neuropsychological problems	Yes (ref. no)	1	1.09 (0.53; 2.23)	0.58 (0.25; 1.35)	1	0.92 (0.44; 1.93)	1.03 (0.50; 2.16)	1	1.40 (0.67; 2.93)	0.90 (0.41; 1.95)
Taking more than 3 prescription drugs/day	Yes (ref. no)	1	0.82 (0.46; 1.46)	0.74 (0.41; 1.33)	1	0.50 * (0.29; 0.89)	0.54 * (0.31; 0.92)	1	1.08 (0.61; 1.91)	0.85 (0.49; 1.49)
No. of medically certified diseases	1 (ref. 0)	1	0.55 (0.28; 1.10)	1.02 (0.52; 2.01)	1	0.58 (0.29; 1.15)	0.51 * (0.27; 0.97)	1	0.70 (0.37; 1.34)	0.56 (0.29; 1.06)
	2–5 (ref. 0)	1	0.99 (0.48; 2.04)	1.71 (0.84; 3.48)	1	0.72 (0.37; 1.39)	0.32 ** (0.16; 0.64)	1	1.29 (0.64; 2.61)	1.14 (0.57; 2.25)

Notes: ^1^ data for nutritional status were collected based on the Mini Nutritional Assessment (MNA^®^) of malnourished (Malnutrition Indicator Score < 17 points), at risk of malnutrition (Malnutrition Indicator Score 17–23.5 points), or normal nutritional status (Malnutrition Indicator Score 24–30 points); ^2^ in the last 3 months due to loss of appetite, digestive problems, chewing or swallowing difficulties, etc.; ^3^ in the last 3 months; ^4^ in comparison with other people of the same age; ORs were adjusted for: age (continuous variable in years), gender, BMI (continuous variable in kg/m^2^), and SESI (continuous variable in points); n—sample size; *p*-value—level of significance assessed with Wald’s test; * *p* < 0.05, ** *p* < 0.01, *** *p* < 0.001.

## Data Availability

Due to ethical restrictions and participant confidentiality, the data cannot be made publicly available. However, data from the ‘ABC of Healthy Eating’ study are available upon request for researchers who meet the criteria for access to confidential data. Data requests can be sent to the ‘ABC of Healthy Eating’ study coordinator (Jadwiga Hamulka).

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
