# Peer review of "Dietary–Physical Activity Patterns in the Health Context of Older Polish Adults: The ‘ABC of Healthy Eating’ Project"

_nutrients, 2022, doi:10.3390/nu14183757_

Round 1
Reviewer 1 Report
This is a valuable study that examined the dietary-physical activity patterns (D-PAPs) in the health context of Polish people aged older than 60 years. The manuscript is well written in an appropriate format for a scientific article. To improve the manuscript further, I hope you consider making the following revisions:
1. Entire manuscript: Since there has been a recommendation against the use of the word "elderly," the term "older adults" may be used instead.
(https://journals.lww.com/jgpt/Fulltext/2011/10000/Use_of_the_Term__Elderly_.1.aspx)
2. Abstract, Lines 89–94, and Figure 1: To further clarify the participants' selection, please indicate the total number of recruited participants from the eight regions and exclusion criteria, etc., according to the Strengthening the Reporting of Observational Studies in Epidemiology (STROBE) statement: guidelines for reporting observational studies (https://www.equator-network.org/reporting-guidelines/strobe/).
3. Abstract: Please add the statistical method (logistic regression analysis) used for the results listed.
4. Introduction, Lines 79–82: The significance of this study would be further enhanced if details of the significance of examining "whole foods’ consumption and [...] dietary patterns" (e.g., specific examples of "some advantages") were included.
5. Tables 2 and 3: The components in the tables would be easier to read if they were left-aligned.
6. Tables 4 and 5: Please provide the actual/true number of participants, as well as the percentage [e.g., number (%)]. Please list the actual/true values even if any P-values were not significant.
7. Table 6: Please add what the "bottom", "middle", and "upper" represent (being tertiles) in a footnote to Table 6.
8. Throughout the entire Discussion section: Even though this study is examining patterns in diet and physical activity, the discussion and conclusion sections are biased toward the diet. I believe it is necessary to mention that the effects of physical activity (physical inactivity).
9. Discussion section, Lines 36–38: As the author understands, since the adjusted analysis is not significant, this statement should be more modestly phrased.
10. Strengths and Limitations, Lines 95–99: Although the participants were selected from eight different regions, there seems to be a bias. Is the percentage of men older than 60 years of age in Poland approximately 13%? Whether the results of this study have generalizability needs to be carefully determined, considering the participation rates from the eight regions.
11. Strengths and Limitations, Lines 99–105: Has the validity of the originally developed Functional Limitation Score (FLS) been confirmed? If it has not been validated, it is unacceptable to highlight it as a strength. In fact, it was unable to detect any difference within the "pro-healthy eating and more-active" group.
12. Discussion section, Lines 107–110: Please provide the basis (a specific formula) for calculating the specific sample size for this study. Is this sample size sufficient for all analyses, not just the main analysis?
13. Strengths and Limitations, Line 105 and Conclusion, Line 126: Since this study did not examine the relationship with "teeth," it is best to refrain from mentioning the status of the dentition or condition of the teeth.
Author Response
Dear Reviewer,
We are excited to re-submit the improved and changed version of our manuscript with a new title: “Dietary-physical activity patterns in the health context of Polish older adults. The ‘ABC of Healthy Eating’ Project”.
We really appreciate all the comments from Reviewer, since they helped us to improve our paper.
We have addressed all issues indicated in the review report, and believe that the new version will meet the journal publication requirements. Thank you very much for a thorough review and insightful feedback. We agree with suggestions and tried to address them accordingly. Please find our responses in the file attached. All changes in the manuscript are highlighted in blue.
Looking forward to hearing from you,
Yours sincerely,
Manuscript authors

Reviewer 2 Report
The authors mentioned that one of the strengths was the creation of the function limitation score based on NMA. However, this was not further discussed.
By using PCA, the authors were able to identify three “dietary-physical activity” patterns. However, it would provide stronger background if the authors can provide some explanation on why or how these food groups are in the same patterns. Especially the “juices, fish, sweetened beverages” patterns.
There is also an inconsistency with the title/intro and the result/discussion of the article. While reading the intro, the author presented that the focus point is on the dietary-physical activity patterns in older adults. However, the result and discussion focus on appetite/ loss of appetite influence on the dietary pattern. Some modification is suggested.
Author Response

(The authors gave the same response as above.)

Round 2
Reviewer 1 Report
Thank you for your polite response to all comments. I only have two minor comments.
Please indicate the total number of invited participants from the 8 regions. If the authors have not surveyed it, please list the population from each of the 8 regions.
The author mentioned that the sample size was adequate if the ratio of a subject to the item was at least 10:1. However, Peduzzi et al (https://doi.org/10.1016/S0895-4356(96)00236-3) indicated that for "event" per variables values of 10 or greater, no major problems occurred. Therefore, it would be appropriate to evaluate the sample size based on the number of participants in each group, rather than all participants. In the case of this study, 230 (23 items × 10) participants need in each group (in the smallest group), so I believe the sample size was insufficient. The author needs to mention that in the limitations.
Author Response
Dear Reviewer,
Thank you very much for a thorough review and insightful feedback of our manuscript with a new title: “Dietary-physical activity patterns in the health context of Polish older adults. The ‘ABC of Healthy Eating’ Project”.
We really appreciate the comments from Reviewer and we have addressed of them, by adding relevant excerpts (explanations) in the manuscript.
Please find our responses in the file attached. All changes in the manuscript are highlighted in green.
Yours sincerely,
Manuscript authors
